# Translation model for CW chord to angle Alpha derived from a Monte-Carlo simulation based on raytracing

Achim Langenbucher[1]*, Nóra Szentmáry[2,3], Alan Cayless[4], Johannes Weisensee[1], Jascha Wendelstein[1,5], Peter Hoffmann[6]

**1** Department of Experimental Ophthalmology, Universität des Saarlandes, Homburg/Saar, Germany, **2** Dr. Rolf M. Schwiete Center for Limbal Stem Cell and Aniridia Research, Universität des Saarlandes, Homburg/Saar, Germany, **3** Department of Ophthalmology, Semmelweis-University, Budapest, Hungary, **4** School of Physical Sciences, The Open University, Milton Keynes, United Kingdom, **5** Medical Faculty, Johannes Kepler University Linz, Linz, Austria, **6** Augen- und Laserklinik Castrop-Rauxel, Castrop-Rauxel, Germany

\* achim.langenbucher@uks.eu

## Abstract

### Background

The Chang-Waring chord is provided by many ophthalmic instruments, but proper interpretation of this chord for use in centring refractive procedures at the cornea is not fully understood. The purpose of this study is to develop a strategy for translating the Chang-Waring chord (position of pupil centre relative to the Purkinje reflex PI) into angle Alpha using raytracing techniques.

### Methods

The retrospective analysis was based on a large dataset of 8959 measurements of 8959 eyes from 1 clinical centre, using the Casia2 anterior segment tomographer. An optical model based on: corneal front and back surface radius Ra and Rp, asphericities Qa and Qp, corneal thickness CCT, anterior chamber depth ACD, and pupil centre position (X-Y position: $Pup_X$ and $Pup_Y$), was defined for each measurement. Using raytracing rays with an incident angle $I_X$ and $I_Y$ the CW chord ($CW_X$ and $CW_Y$) was calculated. Using these data, a multivariable linear model was built up in terms of a Monte-Carlo simulation for a simple translation of incident ray angle to CW chord.

### Results

Raytracing allows for calculation of the CW chord $CW_X$/$CW_Y$ from biometric measures and the incident ray angle $I_X$/$I_Y$. In our dataset mean values of $CW_X$ = 0.32±0.30 mm and $CW_Y$ = -0.10±0.26 mm were derived for a mean incident ray angle (angle Alpha) of $I_X$ = -5.02±1.77˚ and $I_Y$ = 0.01±1.47˚. The raytracing results could be modelled with a linear multivariable model, and the effect sizes for the prediction model for $CW_X$ are identified as Ra, Qa, Rp, CCT, ACD, $Pup_X$, $Pup_Y$, $I_X$, and for $CW_Y$ they are Ra, Rp, $Pup_Y$, and $I_Y$.

**Data Availability Statement:** The data relevant to this study are available from Figshare at https://doi.org/10.6084/m9.figshare.19636881.v1 (https://

figshare.com/articles/dataset/MinimalDataset_
CWchord_Langenbucher_xlsx/19636881).

**Funding:** The author(s) received no specific
funding for this work.

**Competing interests:** The authors have declared
that no competing interests exist.

## Conclusion

Today the CW chord can be directly measured with any biometer, topographer or tomographer. If biometric measures of Ra, Qa, Rp, CCT, ACD, $Pup_X$, $Pup_Y$ are available in addition to the CW chord components $CW_X$ and $CW_Y$, a prediction of angle Alpha is possible using a simple matrix operation.

## Background

The angles Alpha and Kappa have previously been widely discussed for their potential impact on visual performance after cataract surgery and corneal refractive surgery [1]. Especially with premium lenses such as toric, enhanced depth of focus (EDOF) or multifocal lenses [2] or with enhanced excimer or femtosecond laser refractive surgery procedures [3–15] large angles Alpha or Kappa were identified as potential risk factors for an unsatisfactory outcome. In corneal refractive surgery, centration of the corneal ablation pattern can be directly performed based on the corneal centre (limbus centre [16]), pupil centre [17], or the Purkinje reflex PI originated from a coaxial light source in the surgical microscope [5, 12], and the benefits and drawbacks of the different centring strategies has been discussed controversially in many scientific papers. In contrast, in cataract surgery intraoperative centration of the lens implant could be performed with some restrictions [18], but the final lens position is mostly determined by wound healing effects and capsular shrinkage in the postoperative interval [8].

For indication or patient counselling prior to cataract surgery a determination of angles Alpha or Kappa is strictly recommended in the literature, especially when implanting lenses that are known to be sensitive to decentration and tilt. However, it is known that angles Alpha and Kappa cannot be directly determined in a clinical setting as the visual axis and the optical axis or pupillary axis are not properly defined or cannot be measured [12, 19]. In classical schematic model eyes such as the Gullstrand-Emsley or Kooijman eye, all refractive surfaces, either spherical or aspheric, are coaxially aligned and therefore the optical axis passing through all vertices of the surfaces is well defined. However, in those classical model eyes, the visual axis or line of sight coincides with the optical axis as the fovea is assumed to be located at the posterior pole of the eye. In contrast, in modern schematic model eyes such as the Liou-Brennan eye [20], while all refractive surfaces are still coaxially aligned, the fovea is shifted slightly temporally and therefore the incident ray angle is tilted with respect to the optical axis, and in addition the pupil centre is decentred half a millimetre in the nasal direction. While the optical axis can still be assumed as the connecting line between the vertices of all 4 refracting surfaces, the visual axis defined as a connecting line originating from an object point to the object-side nodal point of the eye, and again a connecting line from the image-side nodal point to the fovea, the line of sight defined by the axis connecting the object point and the centre of the entrance pupil (not necessarily hitting the fovea), or the fixation axis connecting the object with the centre of rotation of the eye cannot be determined with optical instruments such as tomographers or biometers. Only the pupillary axis can be easily determined from anterior segment tomography, but this axis intersects neither the object nor the fovea.

The angle Alpha is formed between the optical axis and the visual axis of the eye, and the angle Kappa defines the angle between the pupillary axis and the visual axis measured at the centre of the entrance pupil. As these axes cannot be determined with measurement devices, angles Alpha and Kappa cannot be measured by clinicians.

In 2014 Chang and Waring published a paper discussing this specific issue in detail [21]. In response to the many inconsistencies in the definitions and interpretation of these axes, the

authors recommended the use of measures which can be derived directly from any biometer, topographer or tomographer. These instruments measure distances or surface geometries under patient fixation with a fixation target projected to infinity to avoid instrument myopia. Chang and Waring recommended extracting the outline of the entrance pupil, and the Purkinje reflex PI originated from the corneal front surface. From the relative position of the Purkinje reflex PI and the centre of the pupil they defined the Chang-Waring (CW) chord, replacing the confusing terminology of angles Alpha and Kappa by direct measures from the topographer, tomographer or biometer. However, in the literature there is little information on normative data for CW chord [22, 23], and even more importantly, there is no strategy for interpretation of CW chord or for translating CW chord to angles Alpha or Kappa.

The **purpose of this study** was

- to find a way to translate CW chord to angle Alpha based on biometric data of the anterior segment of the eye,

- to extract the lateral position of the pupil centre and the Purkinje reflex PI using raytracing to determine CW chord,

- to apply this strategy to a large clinical dataset from a modern optical anterior segment tomographer in terms of a Monte-Carlo simulation, and

- to generate a multivariable linear model for a simple application in a clinical setting which translates CW chord to angle Alpha and vice versa.

## Methods

### Dataset for the Monte-Carlo simulation

In total, a dataset with 11,277 measurements (measurements performed between October 2017 and April 2021) from one clinical centre (Augenklinik Castrop, Germany) taken using the Casia2 anterior segment tomographer (Tomey, Nagoya, Japan) was considered for this retrospective study. Duplicate measurements of eyes were already discarded at the time point of data export. Measurements from pseudophakic eyes or in mydriasis and data indexed with situation after refractive surgery, ectatic corneal diseases such as keratoconus or keratoglobus, or other corneal pathologies were omitted from the dataset. The data were transferred to a.csv data table using the data export module of the Casia2 software. Data tables were reduced to the relevant parameters required for our raytracing and analysis, consisting of: laterality (left or right eye), central corneal curvature of the corneal front (Ra) and back (Rp) surface in mm, asphericity of the corneal front (Qa) and back (Qp) surface, central corneal thickness (CCT in mm), external anterior chamber depth (ACD) measured from the corneal front apex to the lens front apex in mm, Pupil size (Pup) in mm, as well as the location of the pupil centre ($Pup_X$ in horizontal and $Pup_Y$ in vertical direction), both in mm.

The data were transferred to Matlab (Matlab 2019b, MathWorks, Natick, USA) for further processing. A waiver was provided for this study by the local ethics committee (Ärztekammer des Saarlandes, 157/21).

### Preprocessing of the data and raytracing

Custom software was written in Matlab using class libraries. By convention we defined a Cartesian coordinate system with an origin at the corneal front apex, X axis to the right, Y axis in the superior direction, and Z axis towards the retina. Fig 1 shows a schematic drawing of the situation of a left eye from above. The optical model is represented by a structure consisting of

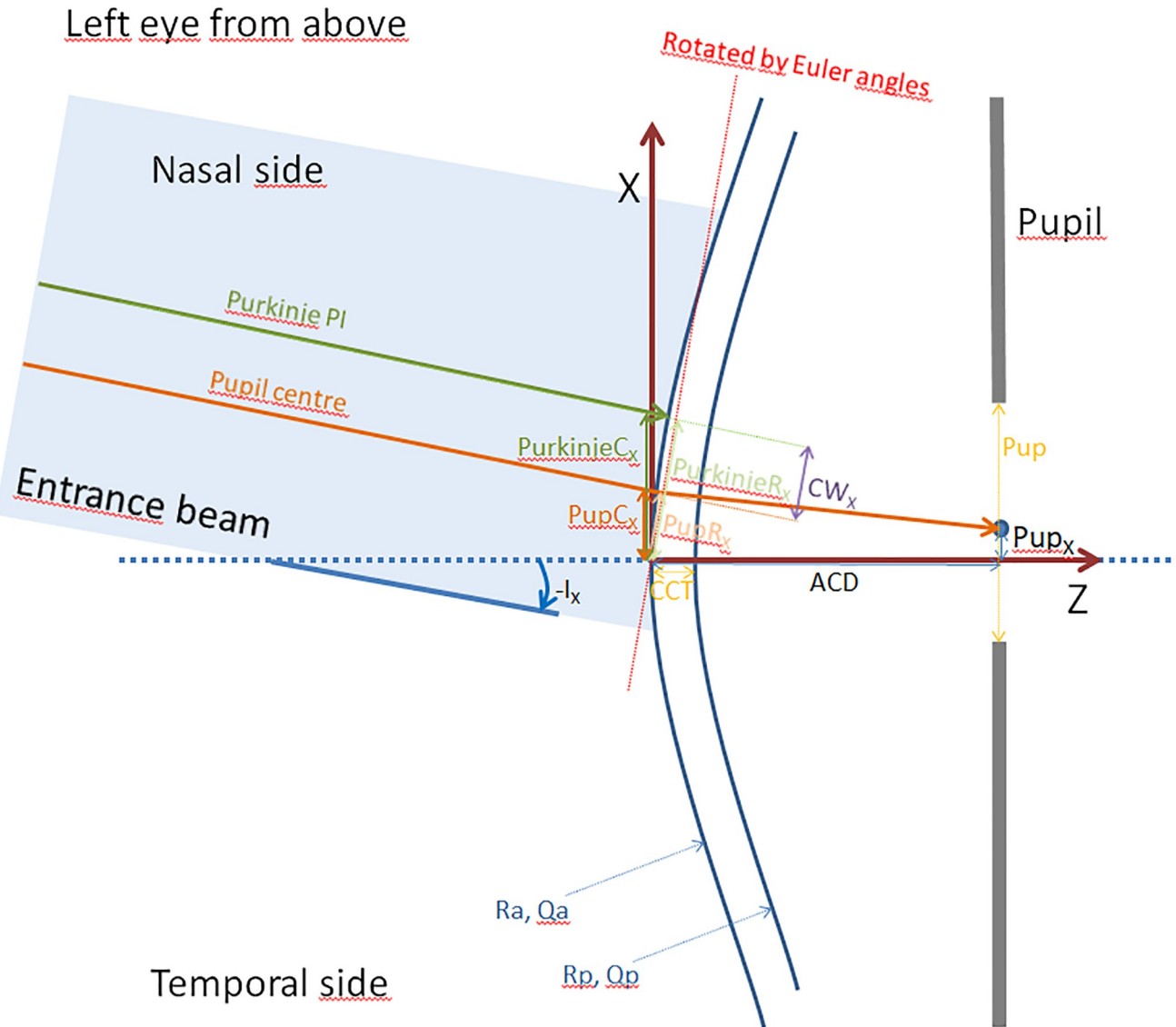

**Fig 1. Schematic drawing of a left eye from above (the vertical coordinates are not shown in the plot). The corneal front and back surface (with radius of curvature Ra and Rp and asphericity Qa and Qp respectively) are coaxially aligned with the Z axis of the coordinate system**. The incident ray is tilted by angle $I_X$ (negative values for $I_X$ as the ray is tilted to nasally). The location of the pupil centre ($Pup_X$) is used to calculate the location of the pupil centre at the corneal apex plane ($PupC_X$). $PurkinjeC_X$ refers to the projection of the Purkinje image PI at the corneal apex plane. Coordinates $PurkinjeC_X$ and $PupC_X$ are rotated to a plane perpendicular to the entrance beam ro read out the coordinates $PurkinjeR_X$ and $PupR_X$ as determined by the tomographer. The Chang Waring chord $CW_X$ is derived from the coordinate points of $PurkinjeR_X$ and $PupR_X$. CCT / ACD / Pup refer to the central corneal thickness / the anterior chamber depth as the distance from the anterior corneal apex to the front apex of the crystalline lens / the pupil size as measured with the Casia 2 tomographer.

a cornea with 2 aspheric surfaces defined by Ra, Qa, Rp, Qp and CCT and an aperture stop with a diameter Pup (Fig 1). Accordingly, the coordinates of the elements in the optical model were X / Y / Z = 0 / 0 / 0 for the corneal front apex, 0 / 0 / CCT for the corneal back apex, and $Pup_X$ / $Pup_Y$ / ACD for the pupil centre (decentration $Pup_X$ in horizontal and $Pup_Y$ in vertical direction), respectively. To avoid unnecessary complexity the front and back surfaces of the cornea were defined to be simple rotationally symmetric aspherical surfaces coaxially aligned on an optical axis coincident with the Z axis. For air / cornea / aqueous humour we used

refractive index values of 1.0 / 1.376 / 1.336 respectively, taken from the Liou-Brennan schematic model eye [20]. This schematic model eye is widely used for optical simulations and ray-tracing purposes, as it appears to be anatomically and physiologically correct and, importantly, it includes non-centred optical elements and a non-coaxial entrance beam thereby considering the decentred location of the fovea in the eye. Without loss of generality, to consider all samples as left eyes, the optical model was flipped horizontally for right eyes, which means that the sign for $Pup_X$ was changed keeping $Pup_Y$ unchanged. Positive values for X and Y refer to the temporal and superior direction.

As the incident ray angle with components $I_X$ / $I_Y$ cannot be measured, and therefore we do not have normative values, we used for the mean value the data from the Liou-Brennan schematic model eye with $mean(I_X) = -5°$ (from the nasal direction, all eyes are considered as left eyes) and $mean(I_Y) = 0°$. Additionally, we added without loss of generality a normal distribution with a standard deviation of 2° and finally discarded values outside the range [-9°; -1°] for $I_X$ and [-4°; 4°] for $I_Y$. With this strategy we considered the eccentric location of the fovea shifted in the temporal direction (positive values of X).

### Calculating the CW chord coordinates

**Pupil centre.**   For initialisation a parametric ray with an incident ray angle $I_X$ / $I_Y$ was projected to the corneal front apex and traced through both corneal surfaces to the pupillary plane. The distance of the ray-pupil intersection was extracted, and using an iterative nonlinear optimisation strategy (Interior Point Method (IPM) [24]) the ray was shifted in X and Y to pass through the pupil centre. At the corneal apex plane the horizontal and vertical coordinates of this ray were $PupC_X$ and $PupC_Y$.

**Purkinje reflex PI.**   For initialisation a parametric ray with an incident ray angle $I_X$ / $I_Y$ was projected to the corneal front apex and refracted by the corneal front surface. From the normalised direction of the ray in front of and behind the corneal front surface, the dot product was analysed. This refers to the cosine of the angle (always in a range from -1 to 1) between the non-refracted and the refracted ray. With an iterative nonlinear optimisation strategy (Interior Point Method [24]), the ray was shifted in X and Y to maximise the dot product (minimise 1-dot product) to identify the ray which is collinear with the surface normal and which is therefore not refracted at the corneal front surface. At the corneal apex plane the horizontal and vertical coordinates of this ray were considered as $PurkinjeC_X$ and $PurkinjeC_Y$.

In a next step the positions of both points ($PupC_X$ / $PupC_Y$ and $PurkinjeC_X$ / $PurkinjeC_Y$) were rotated by Euler angles–$I_X$ / -$I_Y$ with respect to the corneal apex position (Fig 1), to project the pupil centre and Purkinje reflex PI from the Cartesian X / Y / Z to coordinates on a plane perpendicular to the ray ($PupR_X$ / $PupR_Y$ and $PurkinjeR_X$ / $PurkinjeR_Y$). This rotation was performed using quaternion transformation [25].

In a next step we calculated the horizontal and vertical coordinates of CW chord from the offset of the Purkinje PI and the pupil centre by $CW_X = PupR_X—PurkinjeR_X$ and $CW_Y = PupR_Y—PurkinjeR_Y$ (Fig 1).

In a last step we traced a bundle of collimated rays (10,000 equidistant rays) with a diameter of 7 mm through the cornea and the aqueous humour to the pupillary plane. The rays passing through the aperture stop with diameter Pup were marked, and the intersection of these rays with the corneal front apex plane was calculated. A constraining ellipse was derived using eigenvalue decomposition, and the relevant characteristics of this ellipse (centre $EllipseC_X$ and $EllipseC_Y$) together with the major and minor diameter (with angular orientation) were documented. Again, the coordinates of the constraint ellipse were rotated by–$I_X$ / -$I_Y$ using quaternion operation to obtain projections to a plane perpendicular to the incident ray (centre $EllipseR_X$ and $EllipseR_Y$; major diameter $D_{long}@A_{long}$; minor diameter $D_{short}@A_{short}$).

## Setup of the multilinear regression model

A stepwise linear regression [26] was implemented to analyse the relevant effect sizes for a multilinear prediction model for the target parameters $CW_X$ and $CW_Y$ from the potential input parameters Ra, Qa, Rp, Qp, CCT, ACD, $Pup_X$, $Pup_Y$, Pup, $I_X$, and $I_Y$. The stepwise strategy begins with an initial constant model and takes forward and backward steps to add or remove variables, until a stopping criterion is satisfied. As stopping criteria we restricted the number of iterations to a maximum of 100, iteration steps smaller than 10e-9, or improvement of the root mean squared prediction error by less than 10e-12. The tolerance for adding terms was a significance value less than 0.05, and the tolerance for removing terms was a significance value larger equal 0.05.

With the effect sizes identified with this stepwise fit a multivariable linear model was set up to predict the coordinates of the incident ray angle (angle Alpha) to coordinates of CW chord. In addition, for prediction of angle Alpha from the CW chord coordinates, we reversed this linear model to obtain $I_X$, and $I_Y$ from the input variables Ra, Qa, Rp, Qp, CCT, ACD, $Pup_X$, $Pup_Y$, Pup, $CW_X$, and $CW_Y$.

## Results

From the 11,277 measurements exported from the Casia 2 device, totals of 1188 / 789 / 1272 / 365 measurements were indexed as pseudophakic measurements / measurements in mydriasis / eyes with ectatic corneal diseases / incomplete measurements respectively. After quality approval of the dataset and filtering out measurements in pseudophakic eyes, eyes in mydriasis, eyes with ectatic corneal diseases and incomplete data, a final total of N = 8959 measurements (4223 right and 4736 left eyes from 5213 patients) were used for our Monte-Carlo simulation. The process time for extracting the location of the pupil centre, the Purkinje reflex PI, derivation of CW chord, and tracing 10,000 rays through the 8959 optical models took 17,315 seconds (4 hours 49 min) on a standard office PC. Table 1 shows the explorative data for Ra, Qa, Rp, Qp, CCT, ACD, Pup, and the pupil centre location $Pup_X$ and $Pup_Y$ derived from the Casia2 anterior segment OCT, together with the random values defined for the incident ray angle $I_X$ and $I_Y$.

Table 2 shows the descriptive data for the coordinates of the pupil centre $PupC_X$ / $PupC_Y$ and the Purkinje reflex PI ($PurkinjeC_X$ / $Purkinje C_Y$) after raytracing in a plane perpendicular to the incident ray with the origin at the corneal front apex. The total displacement of the pupil centre ($sqrt(PupC_X^2+PupC_Y^2)$) was 0.43±0.19 mm (MEDIAN: 0.41 mm), and for the Purkinje reflex PI it was 0.71±0.23 mm (MEDIAN: 0.71 mm), respectively. These positions are observed from an ophthalmic instrument with coaxial illumination and patient fixation to a far target. The coordinates of CW chord are derived from the difference of $PupC_X$ / $PupC_Y$ and $PurkinjeC_X$, / $Purkinje C_Y$. The data of the constraint ellipse fitted to the rays passing through the pupil

**Table 1. Explorative data extracted from the dataset of the Casia2 anterior segment tomograph.** Ra, Qa, Rp, Qp, CCT, ACD, Pup, $Pup_X$, $Pup_Y$, $I_X$, $I_Y$ refer to the corneal front surface curvature and asphericity, corneal back surface curvature and asphericity, central corneal thickness, anterior chamber depth measured from the corneal front apex, lateral position of the pupil centre in X (positive values in the nasal direction) and Y (positive valuesin the superior direction), and simulated incident ray angle in X and Y. Right eyes were flipped in X. MEAN, SD, MEDIAN, 5% CL, and 95% CL refer to mean value, standard deviation, median, and 90% confidence interval, respectively.

| N = 8959 | Ra in mm | Qa | Rp in mm | Qp | CCT in mm | ACD in mm | Pup in mm | PupX in mm | PupY in mm | $I_X$ in ˚ | $I_Y$ in ˚ |
|---|---|---|---|---|---|---|---|---|---|---|---|
| MEAN | 7.76 | -0.22 | 6.56 | -0.11 | 0.55 | 3.36 | 3.24 | -0.30 | -0.10 | -5.03 | 0.01 |
| SD | 0.28 | 0.13 | 0.25 | 0.11 | 0.04 | 0.40 | 0.81 | 0.21 | 0.19 | 1.77 | 1.47 |
| MEDIAN | 7.74 | -0.22 | 6.56 | -0.10 | 0.55 | 3.37 | 3.31 | -0.30 | -0.10 | -5.03 | 0.02 |
| 5% CL | 7.33 | -0.44 | 6.17 | -0.33 | 0.49 | 2.67 | 2.45 | -0.70 | -0.40 | -7.99 | -2.45 |
| 95% CL | 8.27 | -0.01 | 6.99 | 0.06 | 0.60 | 3.99 | 4.66 | 0.00 | 0.20 | -2.08 | 2.41 |

**Table 2. Explorative data after raytracing and processing: $PupC_X$, $PupC_Y$, $PurkinjeC_X$, Purkinje $C_Y$, $CW_X$, $CW_Y$ refer to the horizontal and vertical coordinates of the pupil centre, Purkinje reflex PI, and Chang-Waring chord as would be noticed by an anterior segment analyser under patient fixation with a far target (projected from Cartesian coordinates X / Y / Z to a plane perpendicular to the incident ray).** $EllipseR_X$, $EllipseR_Y$, $D_{long}$, and $D_{short}$ refer to the horizontal and vertical coordinates of the centre and the long and short diameter of the constraint ellipse of the pupil outline derived from raytracing with a bundle of 10,000 rays. Right eyes were flipped in X. MEAN, SD, MEDIAN, 5% CL, and 95% CL refer to mean value, standard deviation, median, and 90% confidence interval, respectively.

| N = 8959 | $PupC_X$ in mm | $PupC_Y$ in mm | $PurkinjeC_X$ in mm | $PurkinjeC_Y$ in mm | $CW_X$ in mm | $CW_Y$ in mm | $EllipseR_X$ in mm | $EllipseR_Y$ in mm | $D_{long}$ in mm | $D_{short}$ in mm | $D_{long}$ / $D_{short}$ |
|---|---|---|---|---|---|---|---|---|---|---|---|
| MEAN | -0.35 | -0.10 | -0.77 | 0.00 | 0.33 | -0.10 | -0.36 | -0.11 | 3.75 | 3.69 | 1.04 |
| SD | 0.22 | 0.19 | 0.24 | 0.20 | 0.30 | 0.26 | 0.22 | 0.20 | 0.84 | 0.84 | 0.05 |
| MEDIAN | -0.34 | -0.10 | -0.68 | 0.00 | 0.33 | -0.10 | -0.35 | -0.11 | 3.68 | 3.61 | 1.01 |
| 5% CL | -0.73 | -0.41 | -1.08 | -0.33 | -0.16 | -0.54 | -0.74 | -0.40 | 2.56 | 2.49 | 1.00 |
| 95% CL | -0.02 | 0.20 | -0.28 | 0.33 | 0.83 | 0.32 | -0.04 | 0.19 | 4.88 | 4.85 | 1.14 |

in a forward raytracing model characterised by the ellipse centre ($EllipseR_X$, $EllipseR_Y$) and the long ($D_{long}$) and short axis ($D_{short}$) together with the aspect ratio $D_{long}$ / $D_{short}$ are also provided in Table 2. Comparing $D_{long}$ and $D_{short}$ to the pupil diameter measured by the Casia 2 (see also Pup at Table 1) we notice a mean pupil magnification for the entrance pupil of around 15%.

The stepwise fit algorithm which qualifies the input parameters for our linear multivariable model shows that for modelling of the X component of CW chord ($CW_X$), the relevant input values are: Ra (significance level: $P < 1e-9$), Qa ($P = 5.53e-56$), Rp ($P = 0.0016$), CCT ($P = 2.1e-7$), ACD ($P = 0.0394$), $Pup_X$ ($P < 1e-9$), $Pup_Y$ ($P = 0.0019$) and $I_X$ ($P < 1e-9$). Qp, Pup and $I_Y$ did not qualify as input parameters for the model. In contrast, for the Y component of CW chord ($CW_Y$) the relevant input values are: Ra ($P = 0.0900$), Rp ($P = 0.0213$), $Pup_Y$ ($P < 1e-9$) and $I_Y$ ($P < 1e-9$). Qa, Qp, CCT, ACD, $Pup_X$, Pup and $I_X$ did not qualify as input parameters for the model. Fig 2 shows the matrix of grouped scatterplots for the relevant input parameters identified with the stepwise fit algorithm together with the components of the CW chord ($CW_X$ in red, $CW_Y$ in green). The respective histograms are plotted on the diagonal of the matrix. The relevant effect sizes for $CW_X$ are the anterior and posterior corneal curvature Ra and Rp, asphericity of corneal front surface Qa, central corneal thickness CCT, anterior chamber depth ACD, position of the pupil centre $Pup_X$ and $Pup_Y$, and the incident ray angle $I_X$. The relevant effect sizes for $CW_Y$ are identified for anterior and posterior corneal curvature Ra and Rp, position of the pupil centre $Pup_Y$, and incident ray angle $I_Y$. The graph shows that there is a good correlation between Ra and Rp, whereas Qa and Qp are not correlated and show no dependency on Ra or Rp.

The regression model for predicting CW chord ($CW_{XM}$ and $CW_{YM}$) with the regression coefficients composed to a matrix notation is given by the following estimation equation:

$$\begin{bmatrix} CW_{XM} \\ CW_{YM} \end{bmatrix} = \begin{bmatrix} 0.6510 & 0.0870 & -0.0030 & 0.0019 & -0.0753 & 0.0005 & 1.0185 & 0.0016 & -0.1286 & 0 \\ -0.00470 & -0.0002 & 0 & 0.0009 & 0 & 0 & 0 & 1.0199 & 0 & -0.1284 \end{bmatrix} \cdot \begin{bmatrix} 1 \\ R_a \\ Q_a \\ R_p \\ CCT \\ ACD \\ Pup_X \\ Pup_Y \\ I_X \\ I_Y \end{bmatrix},$$

where the non-relevant coefficients are set to zero. The difference in both estimation models ($1^{st}$ and $2^{nd}$ row in the matrix) results from the differences between horizontal and vertical pupil offset ($Pup_X$ and $Pup_Y$) and differences between horizontal and vertical incident ray angles ($I_X$ and $I_Y$). The reversed model for predicting the coordinates of the incident ray angle

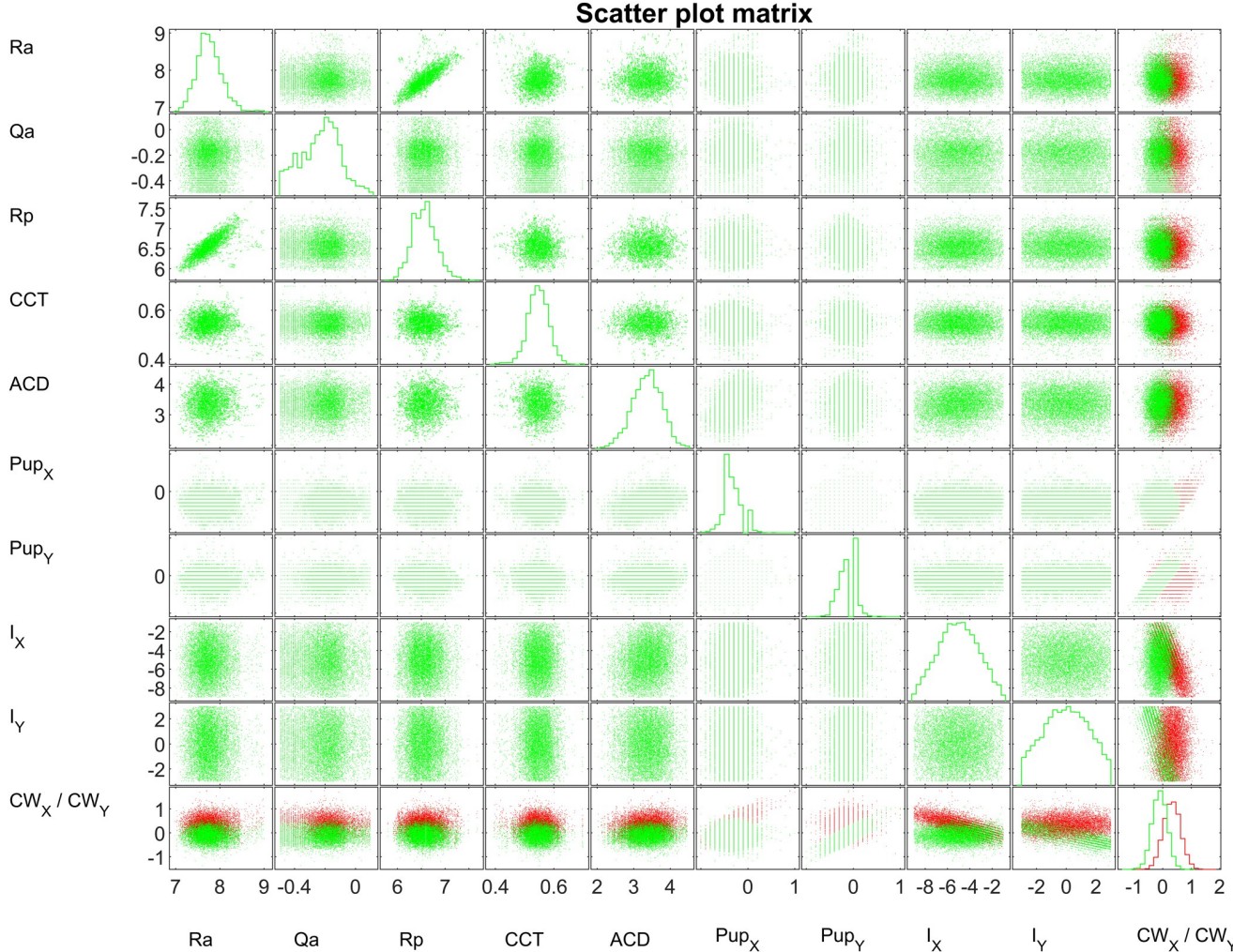

**Fig 2. Matrix of grouped scatterplots for the relevant input parameters of the multivariable linear prediction model and the components of the CW chord (CW$_X$ in red, CW$_Y$ in green).** The respective histograms are plotted on the diagonal of the matrix. The relevant effect sizes for CW$_X$ are identified with a stepwise fit algorithm to anterior and posterior corneal curvature Ra and Rp, asphericity of corneal front surface Qa, central corneal thickness CCT, anterior chamber depth ACD, position of the pupil centre Pup$_X$ and Pup$_Y$, and incident ray angle I$_X$. The relevant effect sizes for CW$_Y$ are identified to anterior and posterior corneal curvature Ra and Rp, position of the pupil centre Pup$_Y$, and incident ray angle I$_Y$. The model is based on a dataset with N = 8959 eye measurements.

(I$_{XM}$ and I$_{YM}$) can be derived after some mathematical transformation as:

$$\begin{bmatrix} I_{XM} \\ I_{YM} \end{bmatrix} = \begin{bmatrix} 5.0659 & 0.6753 & -0.0233 & 0.0149 & -0.5848 & 0.0051 & 7.9067 & 0.0125 & -7.7624 & 0 \\ -0.0357 & -0.0013 & 0 & 0.0071 & 0 & 0 & 0 & 7.9293 & 0 & -7.7738 \end{bmatrix} \cdot \begin{bmatrix} 1 \\ R_a \\ Q_a \\ R_p \\ CCT \\ ACD \\ Pup_X \\ Pup_Y \\ CW_X \\ CW_Y \end{bmatrix}$$

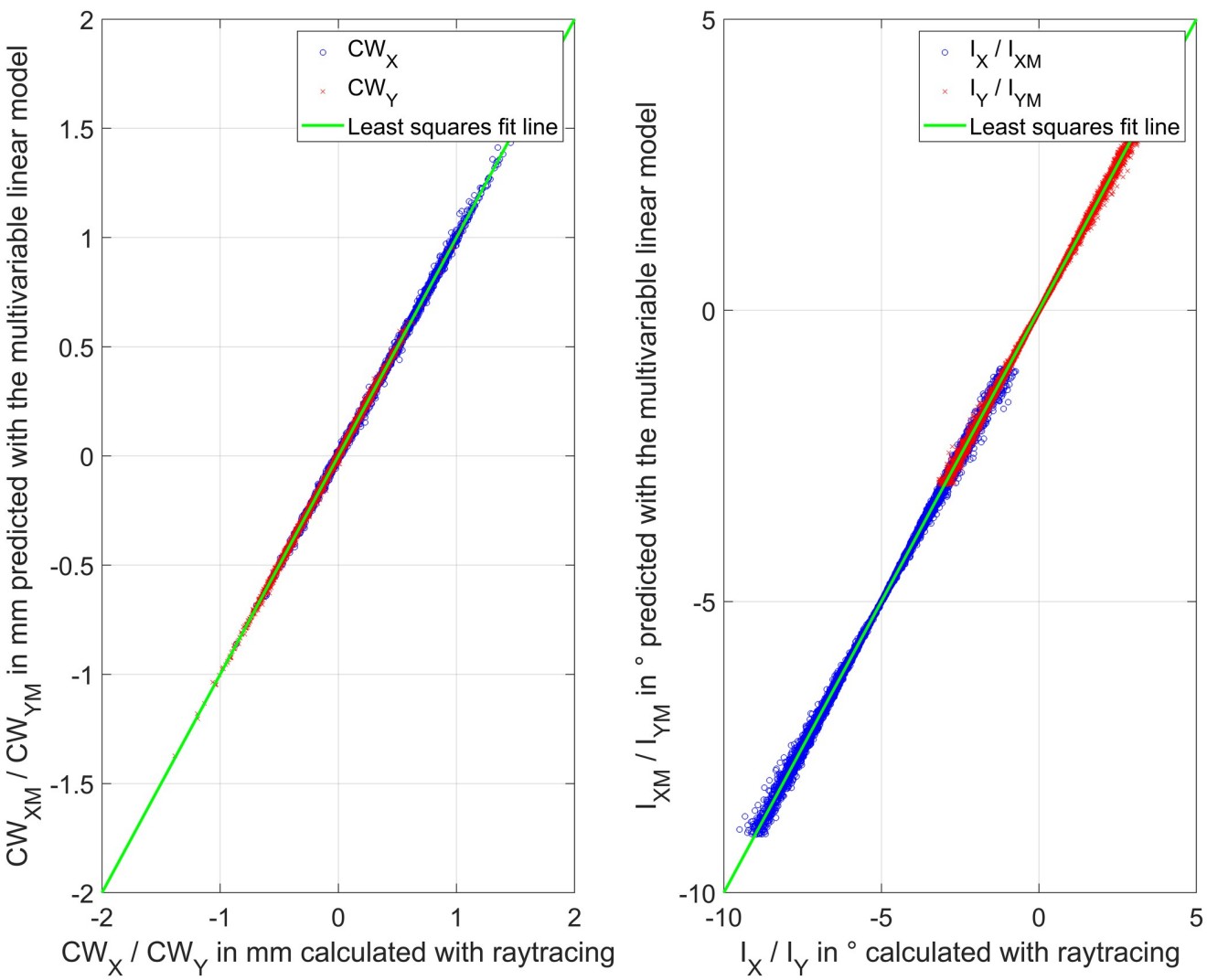

**Fig 3. Performance of the linear models: In the left graph the predicted Chang-Waring chord from the linear multivariable model (X and Y components $CW_{XM}$ and $CW_{YM}$) is plotted versus the respective components derived from raytracing calculations (X and Y components $CW_X$ and $CW_Y$).** In the right graph the reverse model is shown, where the predicted incident ray angle from the linear multivariable model (X and Y components $I_{XM}$ and $I_{YM}$) is plotted versus the respective components derived from raytracing calculations (X and Y components $I_X$ and $I_Y$). The model is based on a dataset with N = 8959 eye measurements.

The performance of both models is shown in Fig 3. On the left side the model predictions $CW_{XM}$ and $CW_{YM}$ are plotted versus the output of the raytracing calculation, and on the right side the respective model predictions $I_{XM}$ and $I_{YM}$ are plotted versus the output of the raytracing calculation.

## Discussion

As previously noted, while there has been much discussion regarding centration of corneal refractive laser procedures and the impact of angle Alpha or Kappa on the outcome of corneal refractive surgery, the impact of intraoperative centration or alignment of the intraocular lens

in cataract surgery is less clear, as the final position of the lens is mostly determined by wound healing effects and capsular shrinkage.

As the angles Alpha and Kappa are not directly measurable in a clinical setting, we wished to explore the use of the CW chord (defined by Chang and Waring as the relative position of the Purkinje reflex PI and the centre of the pupil), as a parameter that can be measured using current instruments such as topographers, tomographers or biometers [27–30]. However, there is to date no calculation scheme which translates the CW chord to angle Alpha or vice versa.

Therefore, in the present paper we set up a raytracing strategy to derive the CW chord from biometric data and the incident ray angle. As discussed in the Methods section, this model was based on selected parameters from a large dataset of measurements from an anterior segment tomographer Casia2, together with additional parameters from the Liou-Brennan model eye. The raytracing model could easily be upgraded to astigmatic or free form surfaces or to non-coaxial arrangement of corneal front and back surface, but the number of effect sizes may increase with the consequence that interpretation of the results might become difficult and the required number of measurements for proper modelling may increase dramatically.

Expressed in the terminology of raytracing, the centre of the entrance pupil is given by the image of the pupil formed by the cornea. This means that we have to find the ray by variation of the lateral position within the ray bundle passing exactly through the centre of the pupil. The Purkinje reflex position PI refers to a location at the corneal front surface where coaxial rays from the fixation target at infinity meet the corneal front surface perpendicularly. In other words, the incident ray is back-reflected; it is collinear with the surface normal (or collinear with the refracted ray). To fulfil this condition we evaluated the dot product between the normalised incident ray and the normalised refracted ray, which equals 1 if it intersects the corneal front surface perpendicularly. Both rays (passing through the pupil centre and hitting the corneal front surface perpendicularly) could be selected from a large number of rays (e.g. 10,000 rays) projected to the cornea considering an incident ray angle by identifying that ray which passes closest the pupil centre or which is refracted the least (shows the largest dot product). In the present study, we wanted to find the ray which passes exactly through the pupil centre and hits the corneal surface exactly perpendicularly. Therefore we implemented a non-linear optimisation strategy which iteratively searches for the ray passing through the pupil centre and passing the corneal front surface perpendicularly by varying the lateral position of the incident ray until our stopping criterion for the iteration was fulfilled. For that purpose we used the Interior Point Method (IPM) [24]. Finally, to obtain the lateral coordinates of the pupil centre and PI as seen by the ophthalmic instrument we projected our locations for pupil centre and Purkinje reflex PI given in Cartesian coordinates to a plane perpendicular to the incident ray (Fig 1). This could be easily performed using quaternion operations based on the classical Euler angles, as well established in computer graphics programming. Finally, we traced 10,000 coaxial rays through our optical model and fitted a constraint ellipse to determine the magnification of the entrance pupil by the cornea and to read out the distortion of the circular pupil to an ellipse resulting from the oblique incident ray angle. However we feel that with a mean aspect ratio of 1.03, distortion of the pupil (3% between long and short axis on average) can be ignored in a clinical setting if the pupil centre has to be identified.

As such raytracing is currently not available in the software tools of ophthalmic instruments, we attempted to model the raytracing results using a linear multivariable prediction algorithm for CW chord. First the relevant effect sizes were determined separately for the horizontal and vertical components $CW_X$ and $CW_Y$. In a second step we calculated the regression coefficients for the relevant effect sizes by minimising the root mean squared prediction error. We discovered that the models for $CW_X$ and $CW_Y$ differ significantly, both in the number of

relevant effect sizes as well as in the regression coefficients. This is mostly due to the fact that the pupil position and the incident ray angles are not identical in X and Y, and that our simple linear setup does not consider non-linear effects with different magnitudes of pupil decentrations and incident ray angles. However, as can be seen from Fig 3, in the parameter space used for our Monte-Carlo simulation the model shows a good performance for the horizontal and vertical component. To provide a calculation for the incident ray angle (angle Alpha) from the biometric measures and CW chord, we also included the reverse prediction model. Both prediction models could be easily implemented in a simple program code, an Excel sheet, or even using the software packages installed on ophthalmic instruments to translate CW chord to angle Alpha or vice versa.

There are some limitations in our study: we restricted the scope to a simple optical model for the cornea with rotationally symmetric aspherical surfaces aligned on a common axis. Further, we are aware that the location of the pupil centre is affected by the pupil size as shown by Erdem et al. [30], Wildenmann and Schaeffel [31], and Yang et al. [32]. For our raytracing setup we used the pupil centre and pupil location as provided from the Casia 2. We did not however perform a series of measurements for each eye with different pupil sizes to assess the respective pupil centre position and the CW chord as a function of pupil size. This was deemed unnecessary as the results from the stepwise fit algorithm confirmed that the pupil size did not act as a relevant parameter in the linear model for translating CW chord to incident ray angle and vice versa. In contrast, when modelling the CW chord from biometric data we feel that the pupil size could be a relevant parameter in the model. And last but not least a translation of CW chord to angle Kappa is in general possible, but requires a more sophisticated optical model as the lens data from the crystalline lens or the intraocular lens after cataract surgery are required to extract the nodal points of the eye. Nevertheless, with the linear regression model the CW chord could easily be translated to angle Alpha if, in addition to $CW_X$ and $CW_Y$, measurements of Ra, Qa, Rp, CCT, ACD, $Pup_X$, $Pup_Y$ are available. Furthermore, if such a conversion is to be applied in the future, the device is a further factor that has to be accounted for. While most biometers and topographers display the apparent CW chord, Scheimpflug tomographers and OCT devices usually display the actual chord μ. These two are not interchangeable, as the apparent CW chord is defined as the chord length between Purkinje-Sanson image I and the apparent pupil centre viewed coaxially from a light source through the cornea, while the actual chord μ displays the actual distance from the visual axis and the actual pupil centre [22].

In **conclusion**, many ophthalmic instruments measure corneal front and back surface curvature and asphericity, central corneal thickness, anterior chamber depth and the location of the pupil in horizontal and vertical direction together with the CW chord defined as the chord between Purkinje reflex PI and the location of the pupil centre. With these data raytracing could be performed to translate CW chord into angle Alpha, or a simplified multivariable regression model as shown in the present paper could be applied.

## Author Contributions

**Conceptualization:** Achim Langenbucher.

**Data curation:** Nóra Szentmáry, Johannes Weisensee, Jascha Wendelstein.

**Formal analysis:** Alan Cayless.

**Investigation:** Achim Langenbucher, Jascha Wendelstein.

**Methodology:** Achim Langenbucher, Johannes Weisensee.

**Project administration:** Peter Hoffmann.

**Resources:** Nóra Szentmáry, Johannes Weisensee, Peter Hoffmann.

**Software:** Achim Langenbucher.

**Supervision:** Nóra Szentmáry, Peter Hoffmann.

**Validation:** Jascha Wendelstein, Peter Hoffmann.

**Visualization:** Alan Cayless, Peter Hoffmann.

**Writing – original draft:** Achim Langenbucher, Alan Cayless.

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
