## [Decision Letter · Decision Letter 0]

8 Feb 2022

PONE-D-21-34567Translation model for CW chord to angle Alpha derived from a Monte-Carlo simulation based on RaytracingPLOS ONE

Dear Dr. Langenbucher,

Thank you for submitting your manuscript to PLOS ONE. After careful consideration, we feel that it has merit but does not fully meet PLOS ONE’s publication plosone@plos.org. Please submit your revised manuscript by Mar 25 2022 11:59PM. If you will need more time than this to complete your revisions, please reply to this message or contact the journal office at plosone@plos.org. Please include the following items when submitting your revised manuscript:A rebuttal letter that responds to each point raised by the academic editor and reviewer(s). You should upload this letter as a separate file labeled 'Response to Reviewers'.A marked-up copy of your manuscript that highlights changes made to the original version. You should upload this as a separate file labeled 'Revised Manuscript with Track Changes'.An unmarked version of your revised paper without tracked changes. You should upload this as a separate file labeled 'Manuscript'.

We look forward to receiving your revised manuscript.

Kind regards,

Paul J Atzberger, Ph.D.

Academic Editor

PLOS ONE

Journal Requirements:

a) Did participants provide their written or verbal informed consent to participate in this study?

3. You indicated that Aa ethics vote was not necessary for this study and that the study was registered with the local Ethics Committee. In light of this statement, please could you indicate within the manuscript text whether the IRB has specifically waived the need for ethics approval for your study.  Please could you also provide confirmation from your institutional review board or research ethics committee (e.g., in the form of a letter or email correspondence) that ethics review was not necessary for this study? Please include a copy of the correspondence as an ""Other"" file."

Reviewers' comments:

Reviewer's Responses to Questions

**Comments to the Author**

1. Is the manuscript technically sound, and do the data support the conclusions?

Reviewer #1: No

Reviewer #2: Yes

Reviewer #3: Yes

2. Has the statistical analysis been performed appropriately and rigorously? 

Reviewer #1: I Don't Know

Reviewer #2: Yes

Reviewer #3: Yes

3. Have the authors made all data underlying the findings in their manuscript fully available?

Reviewer #1: No

Reviewer #2: Yes

Reviewer #3: Yes

4. Is the manuscript presented in an intelligible fashion and written in standard English?

Reviewer #1: Yes

Reviewer #2: Yes

Reviewer #3: Yes

5. Review Comments to the Author

Reviewer #1: Line 1-2: please indicate that this is a retrospective study.

Line 59: please define EDOF

Line 104 - 108:

The purpose of this study is to find a way to translate CW chord to angle alpha based on biometric data of the anterior segment of the eye.

There are many issues with the measurement of angle alpha / kappa. Chang-Warring overcame these issues with the introduction of chord mu / CW-chord. The value of chord mu is readily available in a number of machines. Thus, making life much easier for everybody.

Why do the researchers want to complicate things again by translating CW chord back to angle alpha ???

Line 114 - 116: please indicate the range of dates that the measurements were taken.

Line 117: what were the inclusion criteria?

Line 117: why were measurements from pseudophakic eyes or in mydriasis omitted from the data set?

LIne 123 - 124: Anterior chamber depth (ACD) represents the distance between the corneal endothelium and the anterior capsule of the crystalline lens.

However, the researchers measured the ACD from the corneal front apex to the lens front apex; which is incorrect.

Line 129 - 136: Please provide a schematic drawing / diagram to aid the understanding of the readers.

Line 138- There are a few schematic / model eyes. However the researchers chose to use Liou-Brennan schematic eye. The researchers should provide justification for the choice of their schematic/model eye.

Line 138 - 140: In order to consider all samples as left eyes, the optical model was flipped horizontally for right eyes.

The pupil centre is significantly decentered relative to the corneal centre in the nasal and superior direction.

[Wildenmann U, Schaeffel F. Variations of pupil centration and their effects on video eye tracking. Ophthalmic Physiol Opt. 2013 Nov;33(6):634-41. doi: 10.1111/opo.12086. Epub 2013 Sep 17. Erratum in: Ophthalmic Physiol Opt. 2014 Jan;34(1):123. PMID: 24102513.]

Will flipping the right eye horizontally affect the results of this study?

Why don’t the researchers analyze the data for the right eyes and left eyes separately?

Line 145: Why standard deviation of 2 degree? not 1degree or 3 degree??

line 148 - 178: Please provide a schematic drawing / diagram to aid the understanding of the readers.

line 194: report the number of eyes at each stage of study, eg numbers potentially eligible, examined for eligibility, confirmed eligible, included in this study, and analysed. Give reasons for exclusion at each stage.

Line 270-271: To avoid unnecessary complexity the researchers simplified the front and back surface of the cornea to simple rotationally symmetric aspherical surfaces which are coaxially aligned. How will this simplification affect the outcome of this study?

Line 321: The location of the pupil centre is affected by the pupil size. Yet, the researchers chose to ignore this very very important fact.

How will this affect the validity of the results?

Reviewer #2: The authors describe the prediction of both the Chang Waring chord and angle Alpha from a using both a Monte-Carlo simulation and a multivariate regression model applied to a dataset of 8959 eyes. Data was extracted from from a Casia 2 anterior segment tomographer. They provide a comparison between the CW chord/angle alpha measurements predicted from both models.

Dataset size is sufficient for the described analysis. The models are well described and methodology outlined in adequate detail. The data generated (table 2) appears to have reasonable parameters and the CW chord measurement is consistent with published values from the original authors [1].

References

1) Chang DH, Waring GO 4th. The subject-fixated coaxially sighted corneal light reflex: a clinical marker for centration of refractive treatments and devices. Am J Ophthalmol. 2014 Nov;158(5):863-74. doi: 10.1016/j.ajo.2014.06.028. Epub 2014 Aug 12. PMID: 25127696.

Reviewer #3: Thank you very much for this demanding original work. The potential usefulness of the CW-chord in the planning of refractive procedures and cataract surgery has been presented very well. Your statistical applications are well thought out and implemented with foresight. For the interested reader, who previously had no contact with the topic, the work remains somewhat dry. I would encourage you to create an additional illustration of your design of experiments.

Your discussion is initially redundant and takes up the methodology too intensively again. Here you should avoid unnecessary repetitions and limit yourself to the actual discussion of your methodological/statistical approach and results.

6. PLOS authors have the option to publish the peer review history of their article (what does this mean?). If published, this will include your full peer review and any attached files.

Reviewer #1: No

Reviewer #2: No

Reviewer #3: No

---

## [Author Response · Author response to Decision Letter 0]

1 Mar 2022

Dear Professor Atzberger,

thank you for considering our manuscript entitled ‘Translation model for CW chord to angle Alpha derived from a Monte-Carlo simulation based on Raytracing’ for publication for PlosONE! We found the comments and recommendations of the co-editor and the reviewer very helpful and valuable and we have addressed the issues in the revised version of our manuscript. Our responses to the comments and recommendations of the co-editor and the reviewer are in blue.

Reviewer #1:

 Line 1-2: please indicate that this is a retrospective study.

This study is an evaluation of CW chord using raytracing techniques and a modelling of the results with a linear prediction model, to be used in a clinical environment where raytracing is not available or not practical. Such Monte-Carlo studies are based either on a large dataset (as in this study) or on synthetic data. The term retrospective study might be misleading, as the data are primarily used as a basis for calculating the CW chord using raytracing techniques. We feel that Monte-Carlo simulation as used in the title is the better term. However, for clarity we have added the term ‘retrospective study’ both in the Methods section of the Abstract and in the Methods section of the text.

Line 59: please define EDOF

The reviewer is right! In the revised version we have added ’…enhanced depth of focus (EDOF)…

Line 104 - 108:

The purpose of this study is to find a way to translate CW chord to angle alpha based on biometric data of the anterior segment of the eye.

We thank the reviewer for this idea and we have reformulated the ‘purpose of the study’ accordingly.

There are many issues with the measurement of angle alpha / kappa. Chang-Warring overcame these issues with the introduction of chord mu / CW-chord. The value of chord mu is readily available in a number of machines. Thus, making life much easier for everybody.

The reviewer is right. However, the classical textbooks all describe the angles of the eye are described rather than the CW chord, and also the meaning of CW chord is somewhat different to the angle Alpha. CW chord refers to a projection of the Purkinje reflex PI and the pupil centre, and therefore we require biometric data such as corneal curvature and others to translate CW chord to angle alpha and vice versa.

Why do the researchers want to complicate things again by translating CW chord back to angle alpha ???

The angles of the eye (Alpha, Kappa etc.) are described in all the classical textbooks on ophthalmic optics. The purpose was to develop a concept for a translation of the ‘new’ CW chord and vice versa. What we discovered is that from biometric data of the eye CW chord could be easily predicted from the angle Alpha and vice versa using a simple linear model. As the model performance is surprisingly good such a model could be implemented in several software tools as a good alternative to a raytracing setup.

Line 114 - 116: please indicate the range of dates that the measurements were taken.

We have included the date range where the measurements were taken.

Line 117: what were the inclusion criteria?

Primary measurements on all phakic eyes were included

Line 117: why were measurements from pseudophakic eyes or in mydriasis omitted from the data set?

Pseudophakic eyes were excluded from the study as the CW chord might change in pseudophakic eye due to a potential backward shift of the iris, or a decentration or tilt of the intraocular lens implant. Measurements in mydriasis were excluded as it might be difficult to identify the pupil centre in a pharmacologically dilated pupil. In addition, as the data of the incident ray angle are derived from the Liou Brennan schematic model eye (which describes the situation of the eye without pharmacological stimulation of the pupil size) the mean IX/IY = -5°/0° used as average preset value might be incorrect. This could introduce a systematic shift of the reference point for the respective prediction model for translation of angle Alpha to CW chord and vice versa..

Line 123 - 124: Anterior chamber depth (ACD) represents the distance between the corneal endothelium and the anterior capsule of the crystalline lens. However, the researchers measured the ACD from the corneal front apex to the lens front apex; which is incorrect.

We have used the term "anterior chamber depth (ACD)" to refer to the measurement from the corneal front apex to the lens front apex, as this is the definition used in most biometers. This meaning of ACD is also used in all intraocular lens power calculation concepts. In tomographers, the manufacturers sometimes differentiate between ‘internal’ and ‘external’ anterior chamber depth as measured from the corneal endothelium or the corneal epithelium (Tomey Casia 2, Tomey TMS5); or they use the term ACD (for the measurement from the epithelium) and aqueous depth (AQD, measured from the endothelium, e.g. Haag Streit LenStar; Zeiss IOLMaster 700; Heidelberg engineering Anterion). To clarify our use of the term, we have added 'external' to the definition of ACD in the Methods section.

Line 129 - 136: Please provide a schematic drawing / diagram to aid the understanding of the readers.

We thank the reviewer for this advice! In the revised version of the manuscript we have included a schematic drawing (Figure 1) showing a view of a left eye from above. This drawing explains the situation with the incident ray angle IX, the coordinate system used for raytracing (X, Z), and the coordinates of the projections of the Purkinje PI and pupil centre before(PurkinjeCX and PupCX) and after rotation (PurkinjeRX and PupRX) to a plane perpendicular to the entrance beam using Euler angles.

Line 138- There are a few schematic / model eyes. However the researchers chose to use Liou-Brennan schematic eye. The researchers should provide justification for the choice of their schematic/model eye.

The Liou-Brennan schematic model eye is the most widely used model eye for optical simulations, e.g. in the context of developing new intraocular lens implant or assessing the optical performance. The benefit of this model eye is that that it considers an entrance beam which is not aligned to the ‘optical axis’. With a fully symmetrical model eye and a coaxial entrance beam the Purkinje reflex PI and the projection of the pupil will always be on axis, and the CW chord will always be zero. From the Liou-Brennan schematic model eye we did not use data on the corneal front or back surface curvature, corneal thickness, or the position of the pupil centre in the axial or lateral directions, restricting the use of the data to the refractive index of the cornea and aqueous humour, together with the ‘typical’ angle for the incident ray. We have added some text as justification why this subset of data from the Liou-Brennan model eye was used.

Line 138 - 140: In order to consider all samples as left eyes, the optical model was flipped horizontally for right eyes.

This is correct!

The pupil centre is significantly decentered relative to the corneal centre in the nasal and superior direction.

[Wildenmann U, Schaeffel F. Variations of pupil centration and their effects on video eye tracking. Ophthalmic Physiol Opt. 2013 Nov;33(6):634-41. doi: 10.1111/opo.12086. Epub 2013 Sep 17. Erratum in: Ophthalmic Physiol Opt. 2014 Jan;34(1):123. PMID: 24102513.]

We agree with the reviewer! We have added some text to the paragraph already included in the Discussion section of the original manuscript. In fact, using our stepwise fit algorithm the pupil diameter was found not to have any statistically significant impact in our prediction model for transalating the incident ray angle to CW chord or vice versa. We fully agree with the reviewer that the CW chord itself or the incident ray angle itself would be affected by the pupil size and the respective pupil centre dislocation. However, the pupil size is not relevant for translating the incident ray angle to CW chord or vice versa, as the pupil centre itself was considered in our model.

Will flipping the right eye horizontally affect the results of this study?

When we checked our data prior to analysis we observed that Ra, Qa, Rp, Qp, CCT, ACD and PupY showed no noticeable differences between left and right eyes. Additionally, we determined that the PupX values of left and right eyes were symmetrical about the y-axis, meaning that reversing the sign for all right eyes would produce a distribution identical to that for the left eyes. Together with the identical distributions that we used for IY and the symmetrical distributions that we used for IY for our left and right eyes (with symmetry about the y-axis), we feel that it is justified to flip either all left or all right eyes to get a common description of CW chord from the biometric parameters. Flipping all left eyes to right eyes will not affect our model, as we reversed the sign of all relevant parameters symmetrical with respect to left and right eyes (input parameters IX, PupX, and output parameter CWX).

Why don’t the researchers analyze the data for the right eyes and left eyes separately?

We did consider analyzing left and right eyes separately. However, as we noted that Ra, Qa, Rp, Qp, CCT ACD and PupY showed no noticeable differences and PupX showed symmetrical distributions between left and right eyes, we decided to condense our results into a single model (with description of the model performance and the respective plots) instead of 2 separate models.

Line 145: Why standard deviation of 2 degree? not 1degree or 3 degree??

The standard deviation of 2 degrees was used as an assumption. In general, in a Monte-Carlo simulation as outlined before, either all parameters are synthesized by assumptions on their distributions, or we have a large dataset where most of the parameters with their distributions and interactions can be used and only the missing parameters are synthesized. In this study, we used a large dataset from a clinically established tomographer and made use of all data which could be directly used for our raytracing setup. However, as the incident ray angle cannot be measured by any device, we assumed that on average the incident ray angle matches that of the Liou-Brennan schematic model eye. 

As we expect some variation in the incident ray angle (IX / IY) in the population we added a normally distributed random value with some constrains defined for the lower and upper boundary. One of the most important issues in a Monte-Carlo simulation is that the parameter space is adequately addressed. In fact, the exact distribution and/or range of the parameters used for IX / IY as input parameters are not the most important issues for predicting the corresponding CW chord as response parameter. Ultimately the preset distribution of the incident ray angle (with mean and standard deviation) mostly determines the reference point of our linear model. As we do not see a large scatter in the model performance plot (Figure 3 in the revised version of the manuscript) for small/large incident ray angles (or for small or large CW chord values in the inverse model) we feel that the translation model works correctly for the entire range of incident ray angles (between -1° and -9° in the horizontal and -4° to 4° in the vertical direction). However, the reviewer is correct that we did not find any literature about the variance of incident ray angle, probably mostly due to the fact that this angle of the incident ray cannot be measured. With the boundary conditions for the horizontal incident ray angle [-1° to -9°] we wanted to ensure that the fovea is in all cases located temporally to the symmetry axis. Using smaller or larger standard deviations for the incident angle would most probably result in a slightly higher / lower performance of our linear prediction model of CW chord (slightly less or more scatter in Figure 3 (previously Figure 2)), but less / more predictive value for larger deviations of the incident ray angle from the mean value. However, in general, we would not expect any structural change in our prediction model.

line 148 - 178: Please provide a schematic drawing / diagram to aid the understanding of the readers.

We thank the reviewer for this advice! In the revised version of the manuscript we have included a schematic drawing (Figure 1) depicting a view of a left eye from above. This drawing explains the situation with the incident ray angle IX, and defines the coordinate system used for raytracing (X, Z), as well as the coordinates of the projections of the Purkinje PI and pupil centre both before (PurkinjeCX and PupCX) and after rotation (PurkinjeRX and PupRX) onto a plane perpendicular to the entrance beam using Euler angles.

line 194: report the number of eyes at each stage of study, eg numbers potentially eligible, examined for eligibility, confirmed eligible, included in this study, and analysed. Give reasons for exclusion at each stage.

In total, 11,277 primary eye measurements (one measurement per eye) were exported from the Casia 2 device. At the beginning of the Results section we have now specified as follows: ‘From the 11,277 measurements exported from the Casia 2 device 1188 / 789 / 1272 / 365 measurements were indexed as pseudophakic measurements / measurements in mydriasis / eyes with ectatic corneal diseases / incomplete measurements. After quality approval of the dataset and filtering out measurements in pseudophakic eyes, eyes in mydriasis, eyes with ectatic corneal diseases and incomplete incomplete data, a final total of N=8959 measurements…’

Line 270-271: To avoid unnecessary complexity the researchers simplified the front and back surface of the cornea to simple rotationally symmetric aspherical surfaces which are coaxially aligned. How will this simplification affect the outcome of this study?

In all tomographers currently on the market, the data for corneal front and back surface, corneal thickness, axial and lateral position of the pupil, and pupil size (or the outline data) are referenced to the measurement axis. There are no measurement data that enable us to read out the relative positioning and orientation of the refractive surfaces and the pupil with respect to any independent reference axis (which is not affected by the instrument axis during measurement). We therefore decided to use refractive surfaces aligned to a common axis (ignoring decentration and tilt). In contrast, we allowed for a decentration and size of the pupil (taken from the tomographer data) and used an incident ray angle with respect to the axis (defined by the refractive surfaces) to consider the peripheral location of the fovea. In general, our raytracing setup has the full flexibility to consider any decentration or tilt of refractive surfaces if respective data are available.

Line 321: The location of the pupil centre is affected by the pupil size. Yet, the researchers chose to ignore this very very important fact.

How will this affect the validity of the results?

This might be a misunderstanding! First, we excluded measurements on eyes in mydriasis as we are aware that pupil centre is typically dislocated with pharmacologically dilated pupil (as mentioned above). Second, we used the pupil centre (in axial and lateral position) as well as the pupil size for our raytracing simulation (please see line 124 and lines 132 and 133 of the original version of our manuscript). This means that finally we have for each processed measurement in our dataset also the respective pupil size Pup available. We fitted an ellipse to the projection of the pupil outline (entrance pupil) where the data are shown in Table 2. However, our final analysis showed that the pupil size Pup did not act as a relevant input parameter in our linear model for prediction of CWX or CWY (analysed with the stepwise fit algorithm, Results section lines 213-226). To clarify this we have added in the revised version a sentence for both models which lists the input parameters that did not contribute to the model.

Reviewer #2:

The authors describe the prediction of both the Chang Waring chord and angle Alpha from a using both a Monte-Carlo simulation and a multivariate regression model applied to a dataset of 8959 eyes. Data was extracted from from a Casia 2 anterior segment tomographer. They provide a comparison between the CW chord/angle alpha measurements predicted from both models.

Dataset size is sufficient for the described analysis. 

We thank the reviewer for this comment!

The models are well described and methodology outlined in adequate detail. The data generated (table 2) appears to have reasonable parameters and the CW chord measurement is consistent with published values from the original authors [1].

[1] Chang DH, Waring GO 4th. The subject-fixated coaxially sighted corneal light reflex: a clinical marker for centration of refractive treatments and devices. Am J Ophthalmol. 2014 Nov;158(5):863-74. doi: 10.1016/j.ajo.2014.06.028. Epub 2014 Aug 12. PMID: 25127696.

We thank the reviewer for this favorable comment on our manuscript!

Reviewer #3:

Thank you very much for this demanding original work. The potential usefulness of the CW-chord in the planning of refractive procedures and cataract surgery has been presented very well. 

We thank the reviewer for this positive comment!

Your statistical applications are well thought out and implemented with foresight. For the interested reader, who previously had no contact with the topic, the work remains somewhat dry. I would encourage you to create an additional illustration of your design of experiments.

For a better illustration we have added a drawing defining the coordinate system and showing the different coordinates used to identify the position of the Purkinje PI and the projection of the pupil centre.

Your discussion is initially redundant and takes up the methodology too intensively again. Here you should avoid unnecessary repetitions and limit yourself to the actual discussion of your methodological/statistical approach and results.

Thank you for this advice! We have shortened the Discussion section to avoid unneccesary repetitions in the text, referring back where appropriate to the Methods section.

Thank you for re-considering this manuscript for PlosONE

Achim Langenbucher

---

## [Editor Report · Decision Letter 1]

1 Apr 2022

Translation model for CW chord to angle Alpha derived from a Monte-Carlo simulation based on Raytracing

PONE-D-21-34567R1

Dear Dr. Langenbucher,

We’re pleased to inform you that your manuscript has been judged scientifically suitable for publication and will be formally accepted for publication once it meets all outstanding technical requirements.

Kind regards,

Paul J Atzberger, Ph.D.

Academic Editor

PLOS ONE
---

## [Editor Report · Acceptance letter]

6 May 2022

PONE-D-21-34567R1 

Translation model for CW chord to angle Alpha derived from a Monte-Carlo simulation based on Raytracing 

Dear Dr. Langenbucher:

I'm pleased to inform you that your manuscript has been deemed suitable for publication in PLOS ONE. Congratulations! Your manuscript is now with our production department. 

Kind regards, 

on behalf of

Dr Paul J Atzberger 

Academic Editor

PLOS ONE